# Attribution of Extreme Drought Events and Associated Physical Drivers across Southwest China Using the Budyko Framework

**Xupeng Sun** [1,2], **Jinghan Wang** [1,2], **Mingguo Ma** [1,2] and **Xujun Han** [1,2,*]

1 Chongqing Jinfo Mountain Karst Ecosystem National Observation and Research Station, School of Geographical Sciences, Southwest University, Chongqing 400715, China; giser2020@email.swu.edu.cn (X.S.)
2 Chongqing Engineering Research Center for Remote Sensing Big Data Application, School of Geographical Sciences, Southwest University, Chongqing 400715, China
* Correspondence: hanxujun@swu.edu.cn; Tel.: +86-023-68367339

**Abstract:** Drought is a meteorological phenomenon that negatively impacts agricultural production. In recent years, southwest China has frequently experienced agricultural droughts; these have significantly impacted the economy and the ecological environment. Although several studies have been conducted on agricultural droughts, few have examined the factors driving agricultural droughts from the perspective of water and energy balance. This study aimed to address this gap by utilizing the Standardized Soil Moisture Index (*SSMI*) and the Budyko model to investigate agricultural drought in southwest China. The study identified four areas in Southwest China with a high incidence of agricultural drought from 2000 to 2020. Yunnan and the Sichuan-Chongqing border regions experienced drought in 10% of the months during the study period, while Guangxi and Guizhou had around 8% of months with drought. The droughts in these regions exhibited distinct seasonal characteristics, with Yunnan experiencing significantly higher drought frequency than other periods from January to June, while Guizhou and other areas were prone to severe droughts in summer and autumn. The Budyko model is widely used as the mainstream international framework for studying regional water and energy balance. In this research, the Budyko model was applied to analyze the water and energy balance characteristics in several arid regions of southwest China using drought monitoring data. Results indicate that the water and energy balances in Yunnan and Sichuan-Chongqing are more moisture-constrained, whereas those in Guizhou and Guangxi are relatively stable, suggesting lower susceptibility to extreme droughts. Furthermore, during severe drought periods, evapotranspiration becomes a dominant component of the water cycle, while available water resources such as soil moisture decrease. After comparing the causes of drought and non-drought years, it was found that the average rainfall in southwest China is approximately 30% below normal during drought years, and the temperature is 1–2% higher than normal. These phenomena are most noticeable during the spring and winter months. Additionally, vegetation transpiration is about 10% greater than normal during dry years in Southwest China, and soil evaporation increases by about 5% during the summer and autumn months compared to normal conditions.

**Keywords:** agricultural drought; Budyko; water balance; soil moisture



## 1. Introduction

Drought is a pervasive natural disaster worldwide, with over half of the earth's land surface facing the threat of drought [1]. Unfortunately, the frequency, intensity, and duration of droughts have been on the rise in recent years due to the escalating impacts of climate change [2]. Furthermore, the impact of droughts has a significant cumulative effect, which has emerged as a crucial factor hindering the sustainable development of economies and societies [3]. Of all the sectors impacted by drought, agriculture was the most affected, with direct consequences [4]. Agricultural drought is responsible for more than 50% of the total damage caused by drought [5]. The stability of a society and a country's food security are

closely linked to the agricultural sector. Therefore, it has become an urgent issue to conduct scientific analysis and research on agricultural drought [6,7].

The World Meteorological Organization has classified drought into four distinct categories: meteorological drought, hydrological drought, agricultural drought, and socioeconomic drought [8]. The drought index has become a crucial method for monitoring and evaluating drought [9]. Approximately 55 drought indices are widely used for drought monitoring and analysis, and they are roughly categorized into meteorological, hydrological, and agricultural drought indices [10]. The primary input parameters for meteorological drought indexes are typically precipitation and temperature. Widely used indexes include the Standardized Precipitation Evapotranspiration Index (SPEI) [11], the Palmer Drought Severity Index (PDSI) [12], and the Standardized Precipitation Index (SPI) [13]. The hydrological drought indexes typically use runoff, groundwater, and other core input parameters. Examples include the Surface Water Supply Index (SWSI) [14] and the Stream-flow Drought Index (SDI) [15]. The primary focus of agricultural drought indexes is soil moisture and vegetation ecological data, which are crucial for monitoring and analyzing agricultural droughts. Commonly used agricultural drought indexes include the Standardized Soil Moisture Drought Index (*SSMI*) [16], the Soil Moisture Anomaly Percentage Index (SMAPI) [4], the Soil Moisture Anomaly (SMA) [17], and the Evapotranspiration Deficit Index (ETDI) [18]. *SSMI* and EDTI are frequently utilized for quantitative analysis of agricultural drought in the early stages. For instance, Xu et al. [19] employed *SSMI* to monitor drought conditions in the United States, concluding that *SSMI* is capable of accurately capturing the progression of drought. Narasimhan et al. [20] employed ETDI to examine agricultural drought in Texas and discovered that regional wheat and sorghum crop yields exhibited a strong correlation with ETDI and SMDI (r > 0.75), implying that the developed drought index could serve as an effective tool for monitoring agricultural drought. Additionally, the Normalized Multi-band Drought Index (NMDI) [21] and the vegetation condition index (VCI) [22] are frequently employed to monitor agricultural drought. In this study, we opted to use *SSMI* as the monitoring indicator for agricultural drought and analyze the temporal and spatial variations in southwest China from 2000 to 2020 using SSMI.

Many researchers have conducted studies on monitoring agricultural drought, including assessing its intensity, predicting its frequency, and identifying the drivers behind it. Wilhelmi [23] proposed a spatial, GIS-based approach to assess the vulnerability of Nebraska to agricultural drought. The study assessed the physical and social factors that determine agricultural drought vulnerability and demonstrated that the most vulnerable areas in the study area were non-irrigated drylands and rangelands on sandy soils. YuZhang [24] utilized a meta-Gaussian model to forecast spring and summer agricultural droughts in China, taking into account soil moisture, southern oscillation, and other influencing factors. The findings indicated that the El Niño–Southern Oscillation (ENSO) can offer valuable early warning information for predicting agricultural droughts. Rhee and colleagues (2010) developed a new drought index called the Scaled Drought Condition Index (SDCI) for monitoring agricultural drought using multi-sensor data in both dry and wet regions. The results of their study showed that SDCI was effective in monitoring the spatial and temporal dynamics of agricultural drought in arid and humid areas [25]. Agricultural drought has been the subject of numerous studies conducted in various countries [24,26–29]. However, regional agricultural drought drivers have rarely been analyzed from a water balance perspective.

The Budyko model is widely recognized as the dominant international theoretical framework for investigating the climatic-hydrological interdependencies of watersheds [30]. The classical quantitative theory of the Budyko model is based on the climatic aridity index and the mean hydrological flux of the multi-year average state. Its main objective is to describe the relationship between the regional long-term water balance, which is limited by energy and water [31]. The Budyko model is widely applied in research on regional water balance and other aspects. For example, Maurer and colleagues utilized the Budyko

model to investigate the impact of drought on the water balance in California. The findings revealed that fluctuations in precipitation and temperature across years had a significant influence on absolute runoff changes [32]. Huang [33] assembled the Budyko model with the standardized precipitation index (SPI) and the standardized runoff index (SSI) to analyze the factors influencing the propagation of meteorological drought to hydrological drought in the Weihe River basin, China. The findings indicated that the transmission of meteorological drought to hydrological drought exhibited notable seasonal features, and the duration of the transmission was positively associated with the parameters of the Fu equation of the Budyko model. The Budyko model functions as a conceptual framework that offers a broad comprehension of the connections between regional water and energy balance elements [34]. It achieves this without requiring high-resolution data or numerous parameters. By considering evapotranspiration, it factors in the non-linear correlation between precipitation and available water across diverse climatic circumstances [35]. The applicability of the Budyko model has been enhanced by incorporating parameters that enable adjustments to the deviations resulting from diverse lower bedding surfaces. These parameters have been introduced into the equations governing the Budyko framework.

Southwest China is the main distribution area of the karst landscape in China, and the ecological environment is fragile and vulnerable to extreme disasters [36]. The frequency and intensity of droughts in Southwest China have increased significantly in recent years, leading to adverse effects on the ecological environment, people's lives, and socioeconomic development [37]. The existing studies related to drought in Southwest China mostly focus on drought monitoring, while research on the drivers of agricultural drought from the water balance perspective is relatively limited. Understanding the drivers of drought is crucial for developing effective responses, and investigating the drivers of agricultural drought can provide a theoretical basis for drought prediction and response [38]. This study analyzes the 2000–2020 agricultural drought in Southwest China using the Budyko model combined with the drought index *SSMI* and looks at the impact of different drivers on regional agricultural drought in terms of water and energy balance. The Standardized Soil Moisture Drought Index (*SSMI*) was calculated using GLDAS soil moisture data to assess agricultural drought in the region. The water balance characteristics of Southwest China were explored using the Budyko model, and the changes in water balance components between consecutive drought years in the region and from 2000 to 2020 were compared and analyzed to investigate the impact of different drivers on agricultural drought. The results of this study could provide valuable insights for developing scientific responses to future droughts in Southwest China.

## 2. Methods and Data

### 2.1. Study Area

Southwest China comprises Yunnan Province, Sichuan Province, Guizhou Province, Guangxi Zhuang Autonomous Region, and Chongqing Municipality [39]. The entire study area is 0–7556 m above sea level and covers an area of about 1,594,000 km$^2$, accounting for about 14.2% of China's total area [40]. Southwest China is known for its abundant forest resources, making it a crucial region for forest conservation. According to data from the sixth forest resources inventory, the area has a total forest area of 573,800 km$^2$, with a forest coverage rate of 36% [41]. As shown in Figure 1, there are significant regional climate differences in Southwest China. The Sichuan Basin has a subtropical monsoon humid climate, while Yunnan Province and Guangxi Province have a subtropical monsoon climate and low-latitude plateau climate [42]. There are also small areas of tropical seasonal rainforest climate in the southern end of Southwest China. The region's relatively flat terrain makes it suitable for agricultural development. The combination of low and high latitudes in Southwest China exacerbates the magnitude of climate change, and the abrupt changes in different mountain ranges can have significant impacts on certain areas [43]. Complex climate change worsens the water cycle, increasing extreme weather frequency [44]. Extreme disaster events in the southwest region have had significant impacts on the ecological

environment and social economy, gradually attracting the attention of various sectors of society.

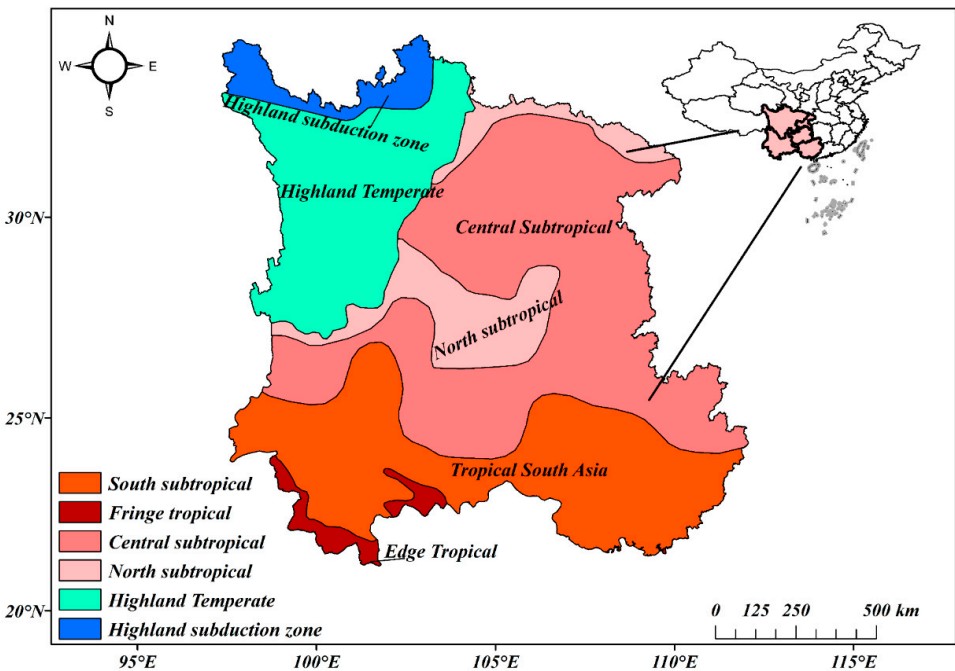

**Figure 1.** The spatial distribution of climate zones in Southwest China.

### 2.2. Data

The data utilized in this study were primarily sourced from the Global Land Data Assimilation System (GLDAS) (https://disc.gsfc.nasa.gov/2105203). GLDAS is a land surface process simulation system that has been collaboratively developed by the National Center for Environmental Prediction (NCEP) and the Goddard Space Flight Center (GSFC) and provides global datasets on various land surface variables [45]. Meteorological, radiation, and other driving data adopted by GLDAS are mainly based on multi-source observations, reanalysis data, and atmospheric assimilation products [46]. The GLDAS assimilation system uses three land surface process models: CLM, NOAH, and the VIC hydrological model [47]. For this study, the research team selected a range of environmental data from the GLDAS dataset, including vegetation evapotranspiration, vegetation transpiration, temperature, precipitation, soil evapotranspiration, soil moisture, and potential evapotranspiration. These variables were chosen specifically for their relevance to the study and have a spatial resolution of 0.25°. Additionally, the temporal resolution of the data is 1 month. The data utilized in this study covers a period from 2000 to 2020.

As a method of fusing models and observation data, land surface data simulations offer a valuable approach to obtaining precise soil moisture and evapotranspiration information that is both temporally and spatially continuous [48]. This approach takes into account the inherent errors present in the observation and background fields while utilizing existing physical mechanisms that contribute to the land surface hydrological cycle [49,50]. The traditional station observation method can provide data with high accuracy. However, the global distribution of stations is relatively uneven compared to land surface process models, and obtaining spatial and temporal continuous data from stations can be challenging [51]. Remote sensing is commonly employed to obtain data at this stage [52]. Remote sensing data have been widely applied in drought monitoring and other research due to the characteristics of remote sensing, including large-area simultaneous observation [53–55]. The optical remote sensing studies in Southwest China have been limited due to the presence of cloudy and foggy environmental conditions in the region. Moreover, microwave remote sensing faces a significant spatial deficit in this area [56–58]. The use of land surface

process models can provide a reliable source of spatial and temporal continuous data for studying drought in Southwest China, as compared to station and remote sensing observation methods, which are limited by factors such as uneven distribution of stations and cloud cover in the region [59]

### 2.3. Agricultural Drought Index SSMI

Agricultural drought refers to the situation where soil moisture content decreases due to external factors, leading to insufficient water absorption by crop roots to compensate for transpiration consumption. This results in an imbalanced water crop balance and abnormal physiological activity [28]. Currently, the primary method used for analyzing and assessing agricultural droughts at the regional level is the use of drought indices. Drought indices such as the Soil Moisture Index (SMI), Vegetation Supply Water Index (VSWI), and Standardized Soil Moisture Index (*SSMI*) are commonly used in agricultural drought research. SMI represents the percentage of actual soil moisture in relation to the field capacity and indicates the soil's relative dryness [60]. Though SMI is suitable for drought monitoring of specific soil types in small-scale areas, it presents certain limitations [? ]. The VSWI is highly susceptible to the impact of cloud and vegetation conditions, and the dense vegetation and frequent cloudy and foggy weather in Southwest China can cause ambiguous VSWI monitoring results [61]. *SSMI* is one of the most direct agricultural drought indices developed and has been validated in different studies [62]. Many studies have confirmed that *SSMI* is a powerful tool for monitoring and evaluating agricultural drought [63–65]. The calculation formula of *SSMI* is

$$SSMI_{i,j} = \frac{\theta_{i,j} - \mu\theta_i}{\sigma\theta_i} \tag{1}$$

where $SSMI_{i,j}$ denotes the *SSMI* for year $j$ and month $i$, $\mu\theta_i$ represents the mean of the long-term series of monthly soil moisture, $\sigma\theta_i$ indicates the standard deviation of the long-time series of soil moisture at the monthly scale, and $\theta_{i,j}$ signifies the mean soil moisture for year $j$ and month $i$ [66]. Combining the *SSMI* results and the relevant government records of droughts in Southwest China [34], we completed a detailed classification of agricultural drought in Southwest China. The classification results are listed in Table 1.

**Table 1.** Agricultural drought severity classification based on the *SSMI*.

| Category | SSMI |
|----------|------|
| Extreme drought | $\leq -2.0$ |
| Severe drought | $-2.0$ to $-1.5$ |
| Moderate drought | $-1.5$ to $-1.0$ |
| Mild drought | $-1.0$ to $-0.5$ |
| No drought | $\geq -0.5$ |

The *SSMI* index defines the average soil moisture value in a given month as the "optimal value" and calculates the deviation of soil moisture from the optimal value to determine the presence of drought. To improve the accuracy of drought severity monitoring, some modifications were made to the *SSMI* index. To calculate *SSMI*, the "normal value" of soil moisture for a single image element was determined by calculating the average value of soil moisture over several years. This "normal value" was then used as a reference in the water balance of the image element. The relationship between the soil moisture content at a given time and the "normal value" was used to determine whether drought occurred. If the soil moisture content at a given time was higher than the "normal value", it was concluded that no drought occurred at that time.

$$SSMI = \begin{cases} SSMI, SM_i < \overline{SM} \\ -0.5, SM_i \geq \overline{SM}, SSMI < -0.5 \end{cases} \tag{2}$$

where $SM_i$ indicates the *i-month* soil moisture, and $\overline{SM}$ denotes the over-year average soil moisture. To reduce the exaggeration of drought monitoring results, the magnitude of the relationship between the multi-year average of soil moisture and the current soil moisture was considered in calculating *SSMI*.

### 2.4. Budyko Model

The focus of this study was to examine the drivers of agricultural drought through the lens of water balance [67,68]. Water balance is the relationship between the water input and output of a system in a particular region at any given time, reflecting the water demand of the system. The Budyko model is a widely accepted theoretical framework for investigating the climate and hydrological interdependence of watersheds on a global scale [30]. The Budyko model is widely used in studies related to regional water balance [69]. The Fu model [70,71] is a modification of the Budyko model; it introduces parameters to adjust for deviations caused by land surface differences and improves the model's performance in the Fu model. The Fu equation is

$$\frac{AET}{P} = \frac{PET}{P} + 1 - \left[1 + \left(\frac{PET}{P}\right)^{\omega}\right]^{\frac{1}{\omega}} \tag{3}$$

where *AET* refers to actual evapotranspiration, *P* represents precipitation, and *PET* signifies potential evapotranspiration. $\omega$ is a parameter that can be influenced by vegetation type, topography, and other factors while indicating the land surface characteristics of the watershed. The Budyko model is explained in greater detail by presenting the basic framework of Budyko and an illustrative diagram of the Fu curve (Figure 2).

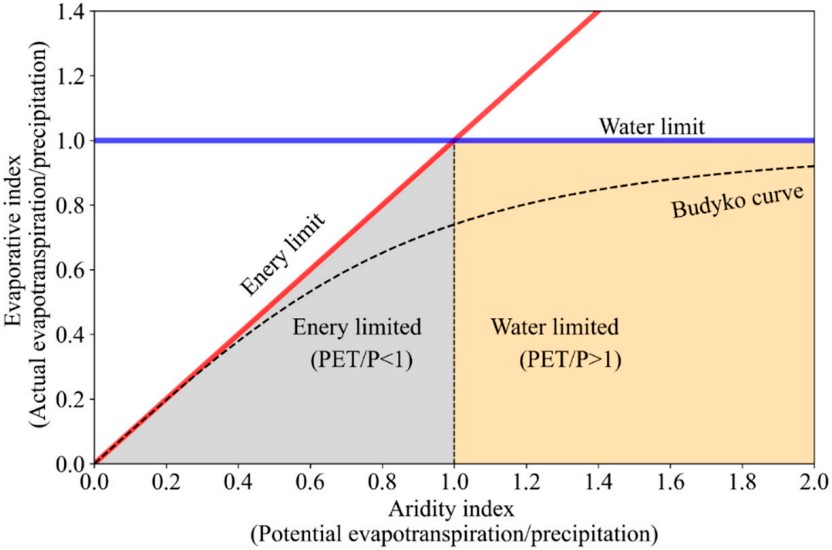

**Figure 2.** Schematic diagram of the Budyko framework. The red line in the diagram represents the energy limitation, and the blue line represents the moisture limitation.

The Budyko model utilizes the ratio of potential evapotranspiration to precipitation as the horizontal axis and the ratio of actual evapotranspiration to precipitation as the vertical axis. All points in the model exist below both the red line (representing energy limitations) and the blue line (representing water limitations). When the regional *PET/P* exceeds 1, the water balance of the area is primarily constrained by moisture, whereas when the regional *PET/P* is below 1, the water balance of the area is largely restricted by energy. A higher dryness index (horizontal axis) signifies a drier region, while a higher evaporative index (vertical axis) indicates a larger proportion of evapotranspiration in water distribution in high regions and a lower proportion of available water resources.

## 3. Results

### 3.1. Results of Agricultural Drought Monitoring in Southwest China

To investigate agricultural drought in Southwest China over time and space, the study utilized *SSMI* to calculate the frequency of moderate-to-severe agricultural droughts in the region between 2000 and 2020. Figure 3 presents the findings, revealing that Yunnan Province, Guizhou Province, Guangxi Province, and the border region of Chongqing City and Sichuan Province have the highest incidence of moderate-to-severe agricultural droughts. Conversely, the frequency of agricultural drought in northeastern Sichuan is relatively low, at approximately 5%, compared to other regions in Southwest China. In the Southwest China, Yunnan Province and the border region of Sichuan and Chongqing experience a high incidence of agricultural drought, with a frequency of approximately 10%. Meanwhile, Guizhou and Guangxi Provinces have around 8%. It is worth noting that the occurrence of drought is slightly more frequent in Yunnan and Sichuan–Chongqing border areas compared to Guizhou and Guangxi.

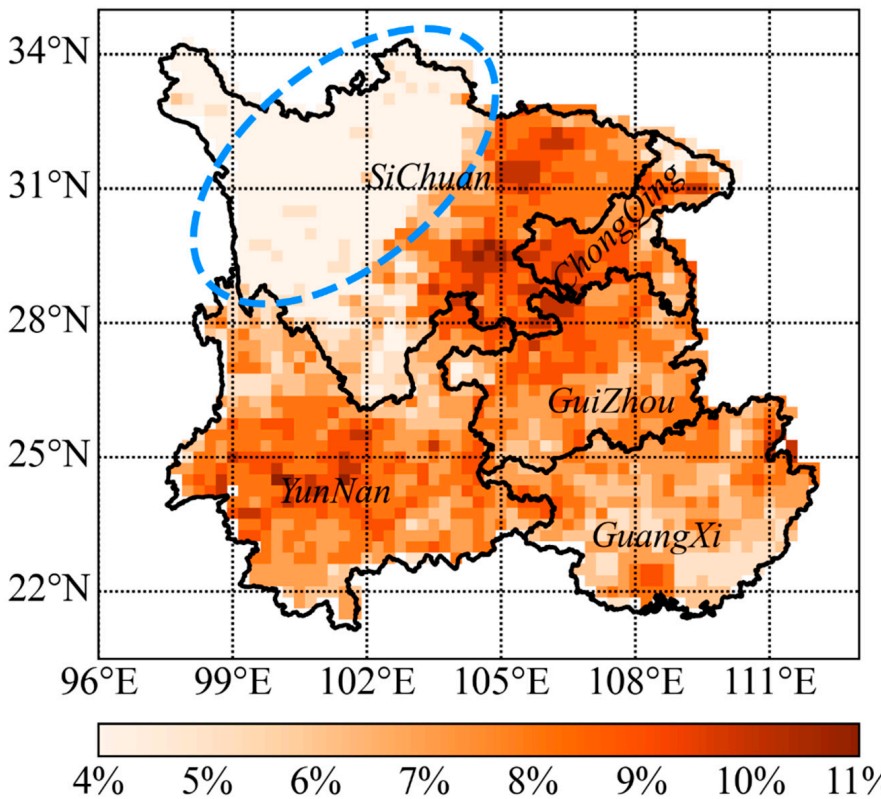

**Figure 3.** The frequency of moderate-to-severe agricultural droughts in Southwest China from 2000 to 2020 was calculated using *SSMI*. The blue line in the figure is the area with a low frequency of agricultural droughts in the southwest region.

Figure 4 exhibits the seasonal drought frequency characteristics of the high agricultural drought-prone regions in southwest China. The frequency of agricultural drought in Yunnan and the Sichuan–Chongqing border region is significantly higher than that in the Guangxi and Guizhou regions, which is consistent with the findings presented in Figure 3. Sub-regional analysis indicates that agricultural droughts in Yunnan Province primarily occur from December to June, with severe types of droughts representing around 8% during this period. The frequency of extreme droughts is approximately 4%. The border area of Sichuan and Chongqing, moderate to severe droughts are prevalent in spring and winter, while extreme droughts are more frequent in summer and autumn (July to October), with the frequency of such extreme droughts at approximately 4%.

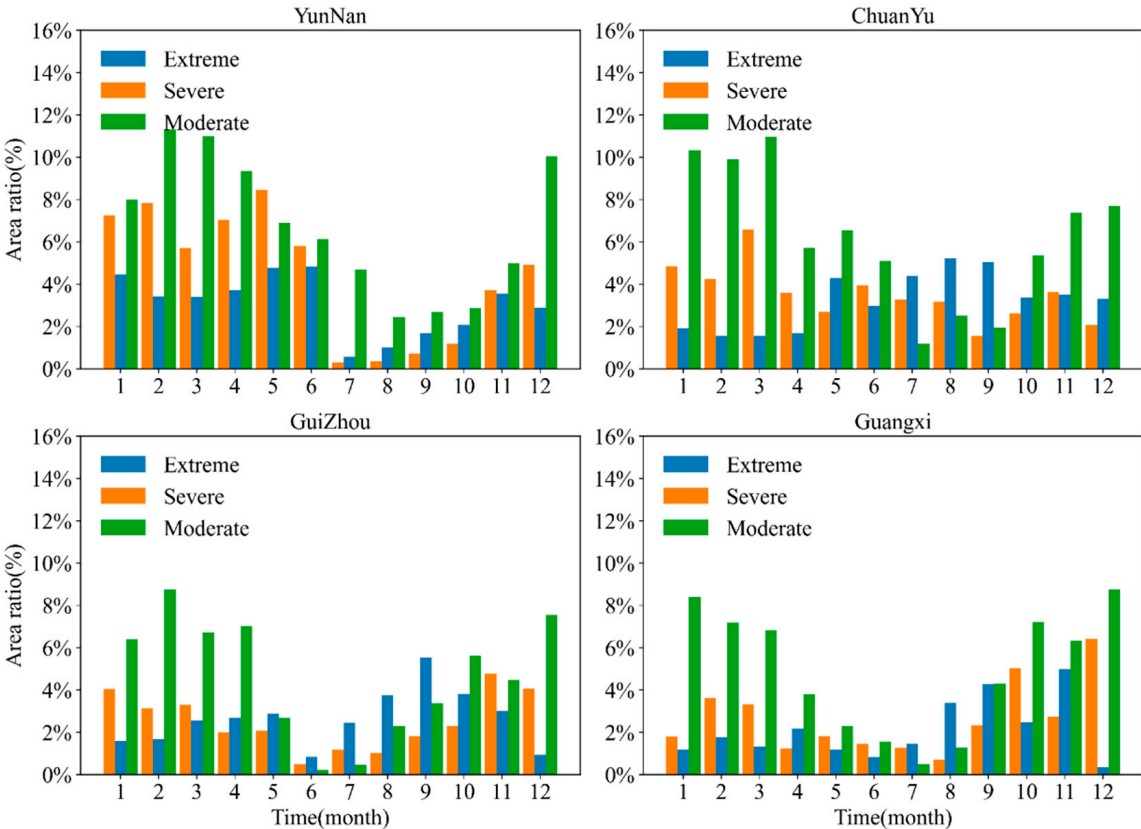

**Figure 4.** The seasonal characteristics of drought frequency in the high agricultural drought-prone regions of Southwest China as calculated by *SSMI*. The vertical coordinates in the figure represent the percentage of drought areas. The vertical coordinates were obtained by calculating the ratio of the cumulative drought area of the region over 16 years to the area of the region. The horizontal coordinates represent the different months.

In comparison to the border areas of Yunnan and Sichuan–Chongqing, the incidence of agricultural drought in Guizhou and Guangxi is significantly lower. The seasonal patterns of agricultural drought in Guizhou and Guangxi provinces are similar. In Guizhou, moderate droughts are more frequent in spring and winter, while severe and extreme droughts occur more often in summer and autumn. In Guangxi, the frequency of moderate droughts is significantly higher in spring and winter compared to other seasons. The frequency of severe and extreme droughts is high from July to November in Guangxi. Throughout all seasons, the frequency of severe droughts is below 8% in both Guizhou and Guangxi provinces.

Figure 5 illustrates the annual-scale *SSMI* variation and the trend of MK variation in four high agricultural drought-prone areas in southwest China. When considering the changes in MK trends, it becomes apparent that the intensity and frequency of agricultural droughts in Southwest China increased significantly between 2000 and 2015. The frequency of droughts weakened significantly after 2015. Moreover, droughts were considerably weaker than the intensity of droughts before 2015. Yunnan Province experienced consecutive drought events from 2009 to 2014. The Sichuan–Chongqing border region experienced severe drought from 2009 to 2014. Guangxi Province experienced consecutive drought events from 2009 to 2012, and the severity of the drought peaked in 2011. Guizhou Province experienced consecutive drought events from 2009 to 2013. In addition to consecutive drought events, Yunnan experienced a severe agricultural drought event in 2019. The Sichuan and Chongqing regions experienced drought severity in 2006 and almost reached the severity of an extreme drought event in 2011. The Guangxi region experienced a severe agricultural drought in 2007.

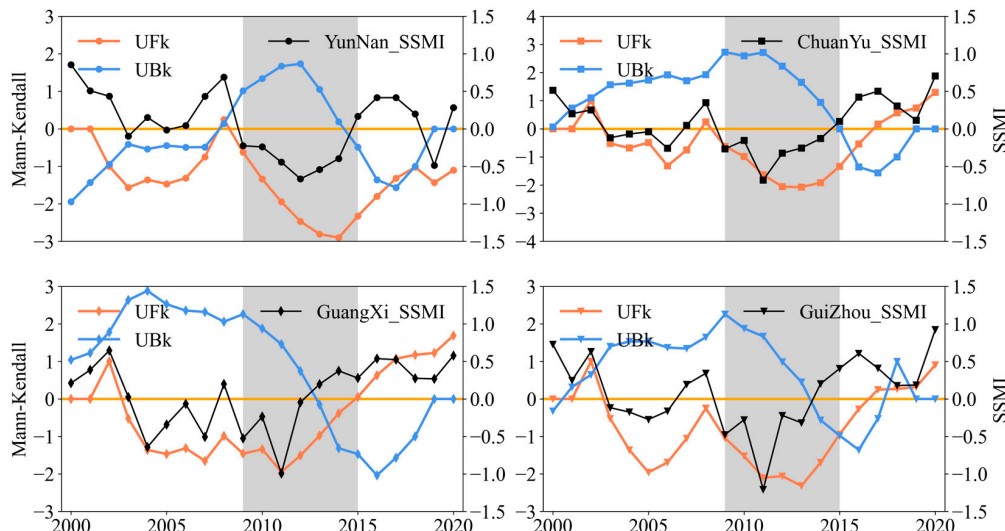

**Figure 5.** *SSMI* variation curves for 2000–2020 for the Southwest region as a whole and for the high drought-prone areas in Southwest China. The grey area in the figure contains the years in which consecutive droughts occurred in Southwest China. The blue and red lines are the UFK and UBK lines of the MK change. The intersection of the two lines represents the inflection point of the trend change.

### 3.2. Budyko Change Trajectory in Southwest China

By integrating GLDAS precipitation and actual and potential evapotranspiration data with the Budyko model, we charted the trajectory of Budyko across various regions of Southwest China, as presented in Figure 6. The results reveal that $PET/P$ in Yunnan and the Sichuan–Chongqing border areas were noticeably higher than those in Guizhou and Guangxi. In simpler terms, the drought intensity was stronger in Yunnan and Sichuan–Chongqing border areas, as compared to Guizhou and Guangxi, consistent with the outcomes of agricultural drought monitoring. The $PET/P$ in Guizhou and Guangxi regions was closer to 1.00, indicating a more stable regional water balance and, hence, a lesser possibility of droughts. The Budyko point was closer to the upper right corner of the coordinate system in Yunnan, Sichuan, and Chongqing regions, implying that these areas were more prone to droughts. Furthermore, the higher the regional evapotranspiration component of the water balance, the lower the percentage of available water resources.

Figure 7 displays the trajectory of single-year Budyko changes across various regions of Southwest China from 2000 to 2020. It is evident that all the Budyko data points in Yunnan Province during this period were situated in the water-limited area. As shown in Figure 5, which reveals annual-scale *SSMI* changes in Yunnan Province, the Budyko point's horizontal coordinate ($PET/P$) and vertical coordinate ($AET/P$) increase as the year becomes drier. For instance, the Budyko point in 2009–2014 was positioned significantly towards the upper right of the coordinate system, indicating an escalation in the degree of drought during those years, a greater proportion of evapotranspiration in the water balance process, and a lower percentage of accessible water resources. The Budyko points for all years between 2000 and 2020 in the Sichuan–Chongqing border area were also distributed in the moisture limitation area, and the Budyko points for the more severe drought years were closer to the upper right of the coordinate system. Between 2000 and 2020, the Budyko points in Guizhou and Guangxi were concentrated around $PET/P$ values of nearly one. In the drier years of these two regions, the Budyko points were skewed towards the upper right of the coordinate system. Nevertheless, the Budyko deviation from the origin was considerably lower in Guizhou and Guangxi than in Yunnan, indicating that these regions were less susceptible to droughts than Yunnan. This also suggests that the water balance in Guizhou and Guangxi was more stable compared to Yunnan.

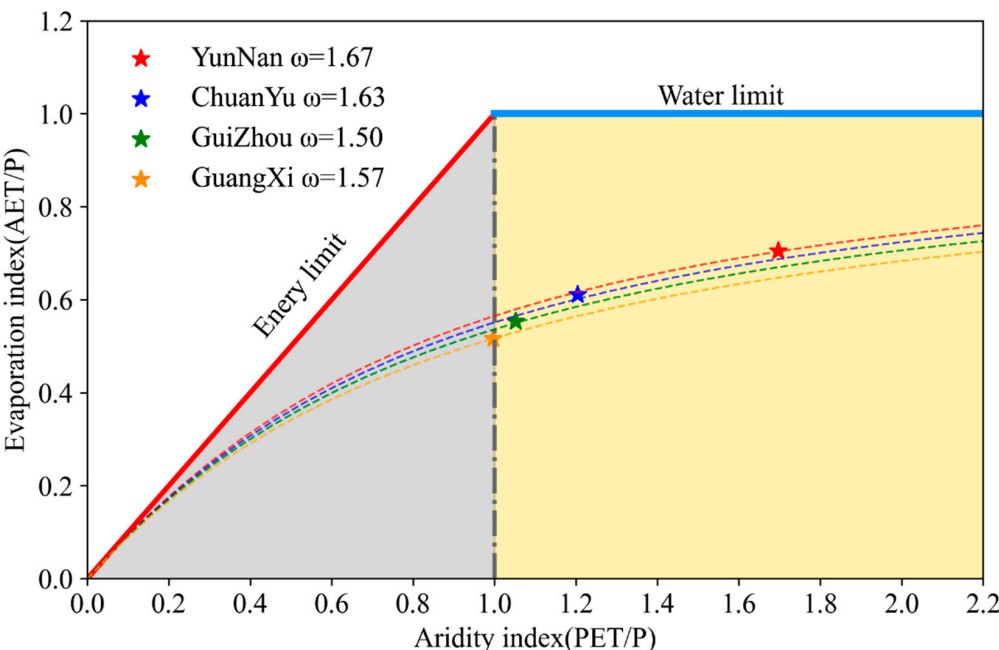

**Figure 6.** The trajectory of Budyko changes in different regions of the southwest. The blue line in the figure represents the water moisture limitation relative to the yellow area. The red line represents the energy limitation close to the grey area.

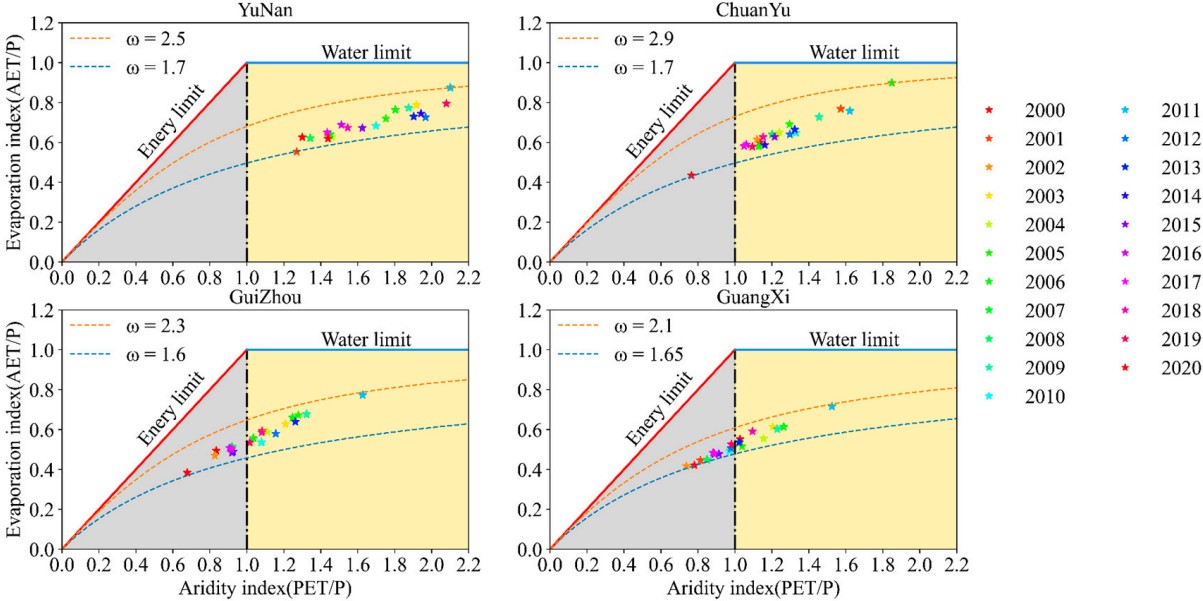

**Figure 7.** Changes in Budyko trajectories in different regions of Southwest China from 2000 to 2020. The red and blue dashed lines in Figure 7 represent the fitted Budyko curves at the maximum and minimum values of $\omega$.

### 3.3. Analysis of Agricultural Drought Drivers

The main components of the water balance include precipitation, soil moisture, evapotranspiration, and runoff [72]. Among them, evapotranspiration can be divided into soil evaporation, vegetation evaporation, and transpiration [73]. Temperature is a significant driving factor in the water cycle [74]. Therefore, this study identified precipitation, evapotranspiration, temperature, and soil moisture as the primary driving factors in investigating the impacts of various factors on agricultural drought. This experiment utilized a data

binning method to analyze the numerical size ranking of the different driven factors. This was done as follows: the time series data for the driven factors were divided into 20 bands based on the 2000–2020 values of the drivers and determining the 5%, 10%, 15% ... 100% of the threshold values.

In this research, the average values of various driving factors during consecutive drought years in Yunnan region were selected and compared with the driving factors from 2000 to 2020. The impact of different driving factors on regional agricultural drought was analyzed from the perspectives of water and energy balance. The results are shown in Figure 8. The Yunnan region experienced a significant decrease in precipitation during consecutive drought years from 2010 to 2015 as compared to the period of 2000–2020. This decrease was particularly evident in the months of January to May, where precipitation levels were consistently at their lowest. As a result, the region experienced a decline in input moisture, which increased the likelihood of agricultural drought events. In terms of temperature, the Yunnan region during drought years recorded temperatures that were approximately 65% of those recorded in the last 20 years. This means that the magnitude of temperature in the drought years in Yunnan is relatively high among all temperatures in the period 2000–2020. This trend was more pronounced from March to May. Additionally, soil moisture levels during consecutive drought years in the Yunnan region were only 10–20% of those recorded in the last 20 years. From an evapotranspiration perspective, the transpiration of vegetation in Yunnan during consecutive dry years was notably higher than in other years. From January to May, the transpiration of vegetation during dry years was almost at its peak compared to the past 20 years, ranging from 65% to 85%. As vegetation evaporation is primarily influenced by precipitation, the trend of vegetation evaporation in Yunnan is in line with the precipitation. During the period from January to May, the average soil evaporation in continuous drought years in Yunnan was significantly lower than in other years. However, from September to November, soil evaporation in dry years in Yunnan was higher than in other years.

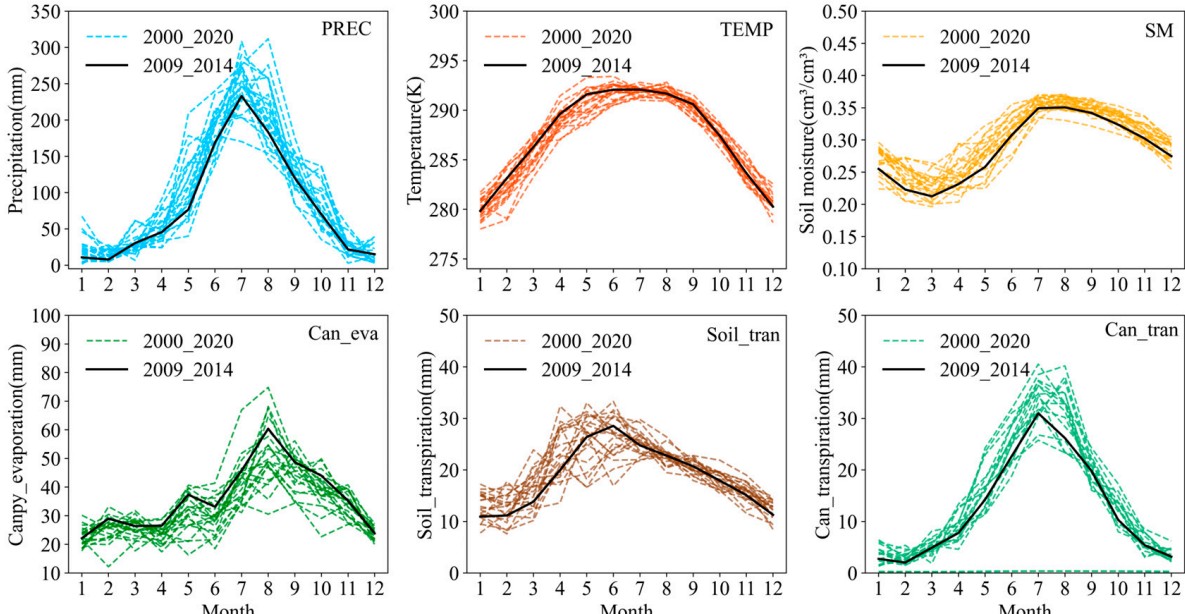

**Figure 8.** Comparative changes in the drivers of agricultural drought in Yunnan during consecutive drought years and from 2000 to 2020. The black line represents the monthly variation of agricultural drought drivers in Yunnan's consecutive drought years (2009–2014). PREC, TEMP, and SM are precipitation, temperature, and soil moisture, respectively. Can_eva, Soil_tran, and Can_tran are vegetation transpiration, soil evaporation, and vegetation transpiration, respectively.

The Sichuan–Chongqing region underwent a series of agricultural droughts from 2009 to 2014. Figure 9 illustrates a comparison between the drivers of agricultural droughts in the region and those of the preceding 20 years. The precipitation in the Sichuan-Chongqing region during the drought years was significantly lower, particularly between July and September. This trend was more pronounced during the continuous drought years. From the perspective of changes in evapotranspiration, vegetation transpiration in the Sichuan-Chongqing region was notably higher during consecutive drought years, while vegetation evaporation was significantly lower in these years compared to others. Soil evaporation during the continuous drought years in the region was markedly lower than that in other years from January to May, reaching nearly the lowest value of soil evaporation observed in the past 20 years. However, soil evaporation from July to November during the continuous drought years in the Sichuan–Chongqing region was significantly higher than in other years, with levels reaching 80–90% of the soil evaporation observed in the past 20 years.

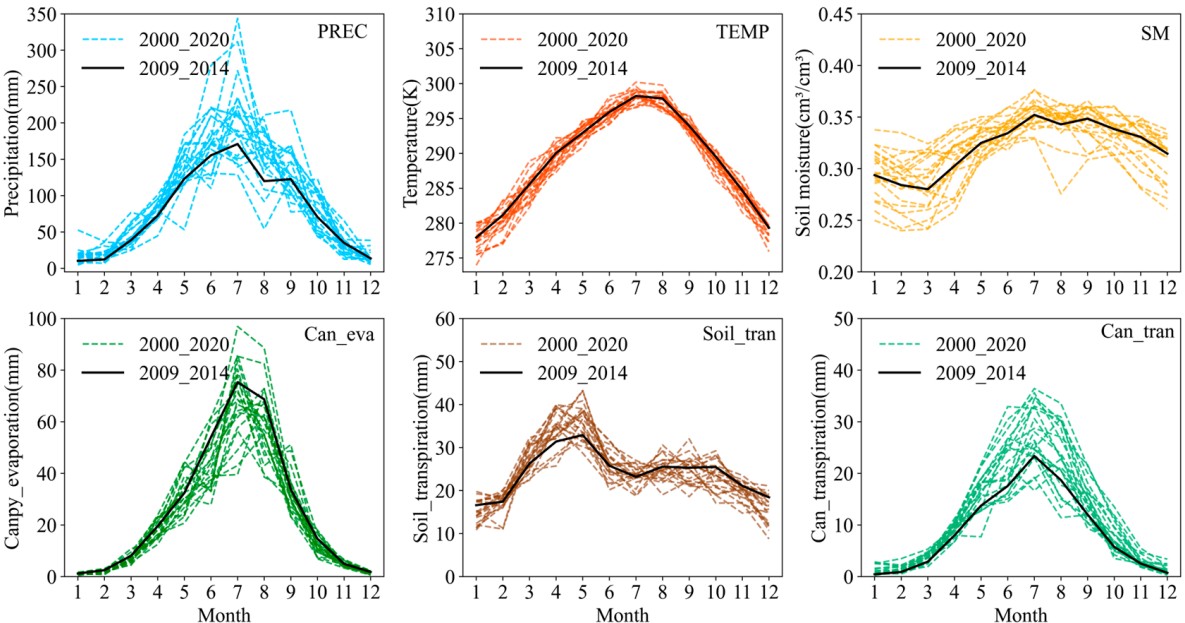

**Figure 9.** Comparative changes in the drivers of agricultural drought in the Sichuan and Chongqing border area during consecutive drought years and from 2000 to 2020. The black line represents the monthly variation of agricultural drought drivers in the Sichuan and Chongqing border area's consecutive drought years (2009–2014). PREC, TEMP, and SM are precipitation, temperature, and soil moisture, respectively. Can_eva, Soil_tran, and Can_tran are vegetation transpiration, soil evaporation, and vegetation transpiration, respectively.

Figures 10 and 11 depict the drivers of agricultural drought in Guizhou and Guangxi during successive drought years and from 2000 to 2020. Agricultural droughts occurred in Guizhou from 2009 to 2013, while Guangxi experienced agricultural droughts for four consecutive years from 2009 to 2012. Figures 10 and 11 illustrate that the average precipitation in consecutive dry years in Guizhou and Guangxi is significantly lower than in other years, especially in summer and autumn, with dry years accounting for about 10–20% of the precipitation in the last 20 years. Regarding temperature, the dry years in Guizhou and Guangxi witnessed temperatures in the upper-middle range (55–60%) over the past 20 years, similar to other regions in Southwest China. In contrast, soil moisture during dry years in Guizhou and Guangxi was significantly lower than in other years. From the perspective of evapotranspiration, the average vegetation transpiration during consecutive dry years in Guizhou and Guangxi was notably high (70–80%) over the past 20 years. Soil evaporation in Guizhou and Guangxi during dry years in January to July was considerably lower than in other years, but soil evaporation during dry years in August to October was

significantly higher than in other years. In terms of vegetation evaporation, it was lower during dry years in Guizhou and Guangxi than in other years.

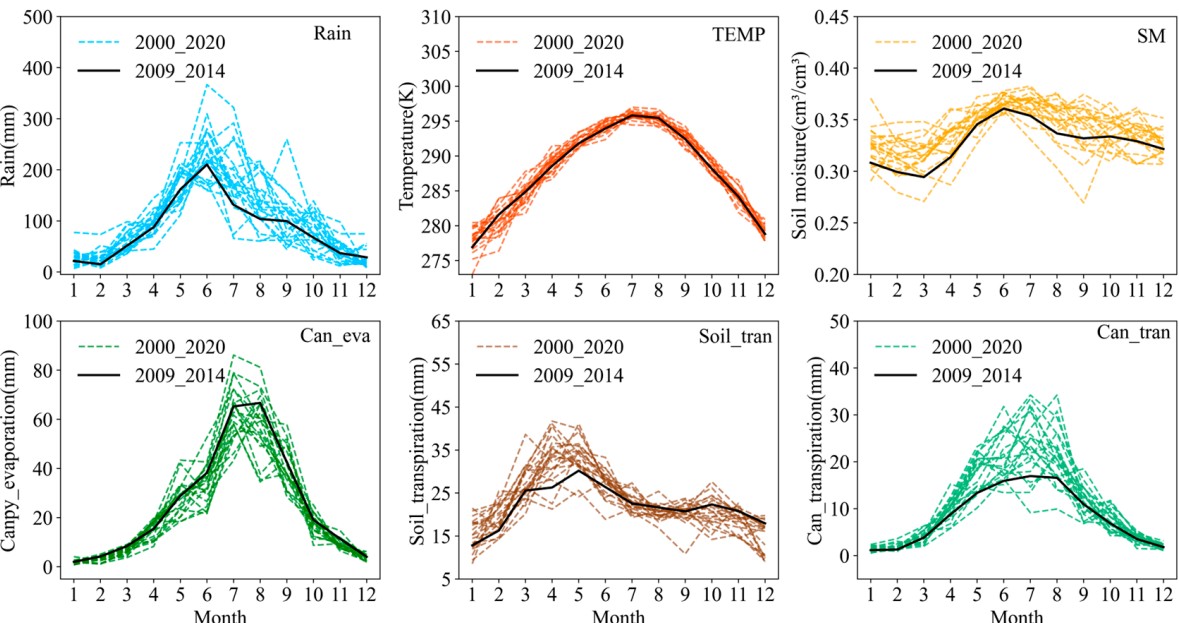

**Figure 10.** Comparative changes in the drivers of agricultural drought in Guizhou during consecutive drought years and from 2000 to 2020. The black line represents the monthly variation of agricultural drought drivers in Guizhou's consecutive drought years (2009–2013). PREC, TEMP, and SM are precipitation, temperature, and soil moisture, respectively. Can_eva, Soil_tran, and Can_tran are vegetation transpiration, soil evaporation, and vegetation transpiration, respectively.

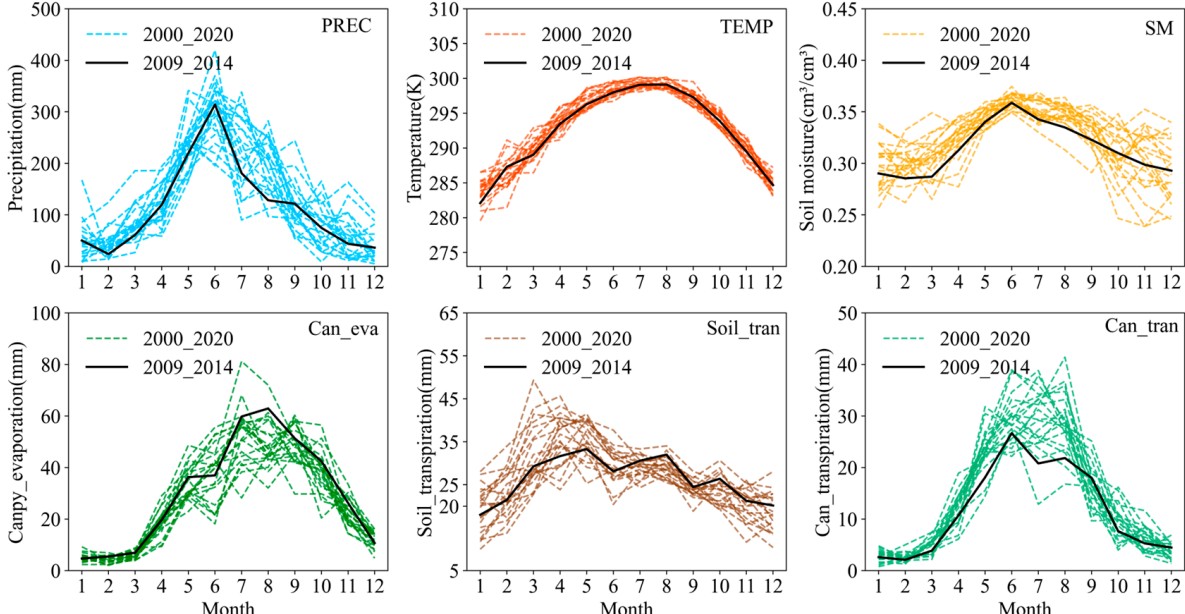

**Figure 11.** Comparative changes in the drivers of agricultural drought in Guangxi during consecutive drought years and from 2000 to 2020. The black line represents the monthly variation of agricultural drought drivers in Guangxi's consecutive drought years (2009–2012). PREC, TEMP, and SM are precipitation, temperature, and soil moisture, respectively. Can_eva, Soil_tran, and Can_tran are vegetation transpiration, soil evaporation, and vegetation transpiration, respectively.

To perform a more quantitative evaluation of the impact of each driving factor on drought, this research measured the degree to which all elements deviated from their normal values in each month. Based on Figure 12, it can be observed that the largest driving factor in causing droughts in Yunnan is the deviation in precipitation. This deviation is especially significant during the spring and winter seasons, reaching up to −51%. The temperature deviation, on the other hand, has a relatively lower impact, with an increase of only about 3% during the high drought season. The skewness in soil moisture can reach a maximum of 7%. The deviation between vegetation transpiration and soil evaporation is second only to precipitation in terms of impact, with the degree of deviation in vegetation transpiration reaching 12% in spring. High deviations in vegetation transpiration can last for approximately three months during summer and autumn. The maximum degree of soil evaporation skewness in Yunnan is about −16%, but there is a positive deviation (around 2–3%) in soil evaporation during summer and autumn in dry years. Supplementary Materials provide data on the deviation of driving factors in other regions and related analysis.

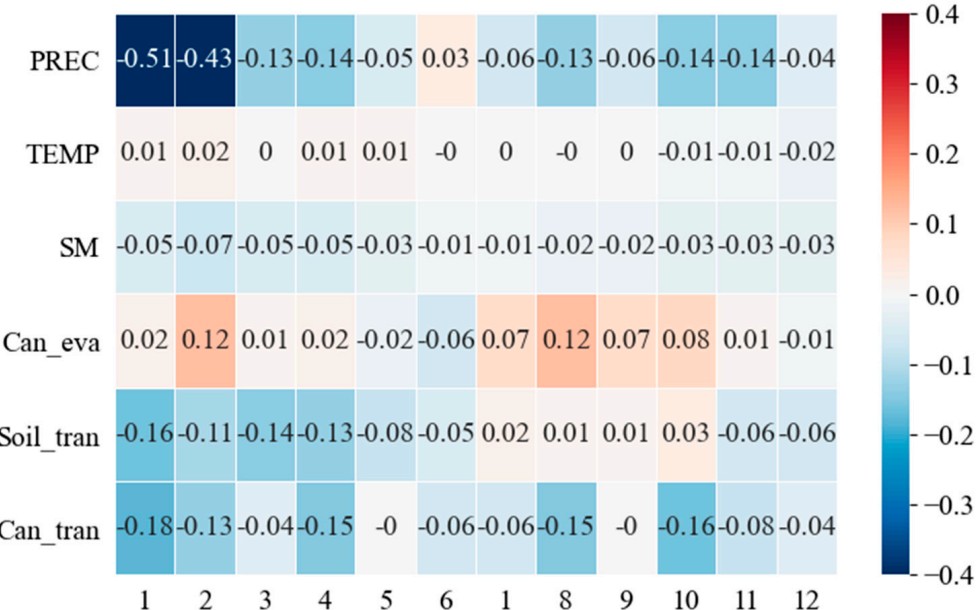

**Figure 12.** Deviation degree of driving factors in different months of drought years in Yunnan.

## 4. Discussion

This research analyzed agricultural drought in Southwest China over the past 20 years by utilizing GLDAS soil moisture data and the *SSMI* drought index. The spatial distribution of agricultural drought was examined, leading to the identification of four high-incidence areas: Yunnan, Guizhou, Guangxi, and the border between Sichuan and Chongqing. Agricultural drought was more frequent in temperate and tropical regions than in boreal regions. The Budyko model was used to analyze water balance characteristics in the high-drought incidence areas, revealing that limited water balance was a significant factor contributing to the frequency of agricultural drought in Yunnan and the Sichuan–Chongqing border areas. In contrast, the water balance in Guangxi and Guizhou was more stable. Finally, the study analyzed the impact of different driving factors on agricultural drought and found that decreased precipitation and increased temperature can cause drought, while vigorous vegetation transpiration and soil evaporation can exacerbate it.

The present research has limitations. Firstly, the agricultural drought index *SSMI* is a single-factor index that is easy to calculate and compare across time and space. However, it does not capture the multi-scale characteristics of agricultural drought, leading to errors in drought monitoring results. In a follow-up study, we will analyze the agricultural drought in Southwest China using multiple indexes. This will help to improve the accuracy of the monitoring results. Furthermore, one of the major shortcomings of the experiment

is that surface runoff and groundwater are not considered in this study. In follow-up research, it is planned to use machine learning in combination with the Budyko model to comprehensively consider the influence of natural and perceived factors on drought. The study mainly focused on long-term monitoring and drivers of agricultural drought, with limited investigation of spatial distribution and changes. Besides its simplicity, the Budyko model has certain limitations. It overlooks a significant portion of the water and energy cycle processes in its computations, which can lead to inaccuracies in the Budyko results. This is due to the fact that it is less complex than modern land surface water and energy balance models. This is due to the fact that it is less complex such models. This issue can be addressed by using a more complete land surface process model (Community Land Model, CLM).

Despite these limitations, the study analyzed the impact of driving factors on agricultural drought in Southwest China from a water and energy balance perspective. It used the agricultural drought monitoring index and Budyko model to quantitatively analyze water balance characteristics in drought-prone areas. The results provide a theoretical foundation for predicting and responding to future agricultural droughts in Southwest China. The methodological framework used in this study can also be applied to other types of droughts for fundamental analysis.

The results of this study on the temporal trends as well as seasonal variations of agricultural droughts in Southwest China are consistent with the results of previous studies and provide side evidence of the applicability of *SSMI* in Southwest China [37,75]. Most previous studies on drought frequency in Southwest China have focused on small areas, such as Yunnan Province or Guizhou Province, while relatively few statistics on drought frequency at large spatial scales have been conducted for the whole of Southwest China [76]. Previous studies on the drivers of drought in Southwest China have mostly been analyzed from the perspective of atmospheric circulation (El Niño, Southern Oscillation, etc.) [77], but this study analyzes the driving mechanisms of agricultural drought in Southwest China from the perspective of water balance. The present study also finds that agricultural drought in Southwest China is not only related to precipitation and temperature, but also to regional evapotranspiration. The results of the current study are consistent with those of Mas et al., validating the plausibility of our results [78,79]. From the perspective of the physical mechanism of drought, the decrease of rainfall and the increase of evapotranspiration may be important inducing factors for the occurrence of drought [80–82]. In addition, human activities have both increased and alleviated the drought [80].

The relationship between rainfall and evapotranspiration and the calculation of surface parameters have been the focus of most previous studies on Budyko. For example, Gerrits [81] used the Budyko model to analyze the relationship between rainfall and evapotranspiration. Bai [82] used a multiple linear regression model and artificial neural network model to calculate Budyko model parameters for the Chinese region. This study applies the Budyko model to the study of drought drivers. It quantifies the drivers of drought in Southwest China by combining water balance principles. This study analyzes the impact of evapotranspiration components on drought, unlike previous studies [34].

## 5. Conclusions

The severity, coverage, and duration of agricultural drought in Southwest China from 2005 to 2020 exhibited obvious spatiotemporal characteristics. The severity of drought in the agricultural sector in Southwest China from 2005 to 2014 showed a significant increase, while the frequency and duration of agricultural drought in the region decreased significantly from 2015 to 2020. Based on the frequency of drought, four high-frequency regions were identified in Southwest China, namely Yunnan, the Sichuan–Chongqing border region, Guangxi, and Guizhou. The frequency of moderate to severe drought in the northwestern Sichuan region was 4–5%, significantly lower than in other regions. Overall, agricultural drought in Southwest China showed a spatial pattern of decreasing severity from the southwest to the northeast. The four high-frequency regions of agricultural

drought in Southwest China exhibited obvious seasonal characteristics, with Yunnan being prone to drought in the spring and winter seasons; the frequency of extreme drought in the winter and spring seasons was around 10%. The Sichuan–Chongqing border region, Guizhou, and Guangxi were prone to moderate drought in the spring and winter seasons, and severe and extreme droughts in the summer and autumn seasons. The frequency of extreme drought in the summer and autumn seasons in the Sichuan-Chongqing border region was about 6%, while that in Guangxi and Guizhou was about 4%.

The Budyko model was utilized to analyze the water balance characteristics of the four drought-prone regions. The results showed that the Budyko points in Yunnan and the Sichuan–Chongqing border areas were located more towards the upper right of the coordinate system (Figures 6 and 7), indicating that water availability was more limited in these areas. Concurrently, the ratio of potential evapotranspiration to precipitation in Guizhou and Guangxi was closer to 1.00. In other words, the water balance in Guangxi and Guizhou was more stable, which also explains why droughts occur more frequently in Yunnan and the Sichuan–Chongqing region than in Guangxi and Guizhou. By analyzing the results of single-year regional Budyko trajectories with regional drought trends, it was found that the more severe the agricultural drought, the higher the percentage of evapotranspiration and the lower the percentage of available water in the components of the water balance. This suggests that the soil moisture is lower in severe drought conditions.

By comparing the changes in driving factors between consecutive drought years and normal years, it was found that in the southwest region of China, consecutive drought years had significantly lower rainfall and higher temperatures. In addition, during the spring and winter seasons of drought years, vegetation transpiration was very active, while during the summer and autumn seasons, vegetation transpiration and soil evaporation were significantly higher than in non-drought years.

**Supplementary Materials:** The following supporting information can be downloaded at: https://www.mdpi.com/article/10.3390/rs15112702/s1, Figure S1: Deviation degree of driving factors in different months of drought years in Chuanyu; Figure S2: Deviation degree of driving factors in different months of drought years in Guizhou; Figure S3: Deviation degree of driving factors in different months of drought years in Guangxi.

**Author Contributions:** X.S.: methodology, investigation, formal analysis, data curation, conceptualization, writing—original draft, writing—review and editing. J.W.: writing—review and editing. M.M.: conceptualization, software, writing—review and editing, visualization. X.H.: visualization, writing—review and editing, funding acquisition. All authors have read and agreed to the published version of the manuscript.

**Funding:** This work was jointly supported by the National Natural Science Foundation of China (NSFC) project [grant numbers: 4112300101, 41830648, 41771361].

**Data Availability Statement:** The data used in this paper are provided by GLDAS official website (https://ldas.gsfc.nasa.gov/gldas/).

**Acknowledgments:** The data on soil moisture and evapotranspiration used in this study are from the Global Land Data Assimilation System (GLDAS), and the statistical year data used are from the Statistical Yearbook of the Chinese Government. We sincerely thank the anonymous reviewers.

**Conflicts of Interest:** The authors declare no conflict of interest.

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
