# Peer review of "Attribution of Extreme Drought Events and Associated Physical Drivers across Southwest China Using the Budyko Framework"

_remotesensing, doi:10.3390/rs15112702_

Round 1

Reviewer 1 Report (Previous Reviewer 1)

Following previous two round revisions I'm now happy with authors effort.

A minor comment is that the manuscript shall be converted in MDPI format using numbered references.

Author Response

Thank you very much for your affirmation of us, and your opinion is very helpful to us, we have adjusted the format of references in time.

Reviewer 2 Report (Previous Reviewer 2)

no comments

Author Response

Thank you very much for your affirmation of our work!

Reviewer 3 Report (New Reviewer)

COMMENT TO AUTHORS

Review for Characteristics and Drivers of Extreme Drought in Southwest China under Budyko Framework, by SUN et al. (2023).

This manuscript attempts to examine the Characteristics and Drivers of Extreme Drought in Southwest China under the Budyko Framework. Extreme climate events continue to pause threats and devastating impacts to society, economy, and ecosystem across many regions globally. A comprehensive understanding of their development and occurrences, with key mechanisms initiating and amplifying them, remains a relevant approach. From this point of view, examining meteorological factors associated with the recent occurrence of compound dry and wet extremes over vulnerable regions is of great importance and relevant information to the scientific community. Overall, the study is interesting and well-written. However, major flaws exist in the present research approach and the results reported in the current version manuscript. The reviewer therefore would like to recommend this manuscript be returned to the authors for MAJOR REVISIONS as suggested below before a revised manuscript may be re-submitted for publication. My comments are given below;

Title# I think the title reflects the research objectives and seems inappropriate. It is suggested to make it more clear and suitable according to the work done in this manuscript.

Abstract# In the abstract the main result findings are lacking, instead of emphasizing motivation and even the research objective is missing. Thus, more results need to add to the abstract with a proper narrative.

In the Introduction: The authors should point out the research goal of this manuscript and emphasize its research significance. Moreover, the authors should introduce the implications of previous related published research from a broader perspective.

https://www.mdpi.com/2072-4292/15/6/1680; https://www.mdpi.com/2072-4292/13/16/3294

https://www.mdpi.com/2072-4292/13/11/2059

What are the main aims and motivations for using different reanalysis datasets instead of observed datasets, which is unclear and even not seen as the scientific reason in the entire manuscript?

I feel the authors need to provide some discussion on the limitation of the study in the discussion section and what needs to be done to address those. For example; a comprehensive analysis of climate sensitivity, given the datasets (that are used in the study) has a significant uncertainty amongst their models. Although I see authors have mentioned this in their manuscript, it still needs more elaborated discussion. Finally, discuss some lights on the physical mechanism on which the dryness/wetness is increasing over the regions much-with the help of some recent literature.

https://doi.org/10.1016/j.scitotenv.2021.148162; https://www.mdpi.com/2072-4292/15/6/1484

https://onlinelibrary.wiley.com/doi/10.1002/joc.7419; https://www.mdpi.com/2073-4441/13/5/729

Conclusions are not clarified. Besides, these findings cannot be well supported by the data and the results.

Minor editing of English language required

Author Response

Response to reviewer

Dear reviewer:

Q1: Title# I think the title reflects the research objectives and seems inappropriate. It is suggested to make it clearer and more suitable according to the work done in this manuscript.

Response:

Thank you very much for your comments, they have been very helpful and we have made some changes to the title of the article.

“Characterizing Extreme Drought and Identifying Drivers in Southwest China using the Budyko Framework”

Q2: Abstract# In the abstract the main result findings are lacking, instead of emphasizing motivation and even the research objective is missing. Thus, more results need to add to the abstract with a proper narrative.

Response:

Thank you very much for your comments, they have been very helpful and we have made some changes to the title of the abstract.

  “Agricultural drought is the most devastating meteorological calamity that negatively impacts agricultural production. In recent years, Southwest China has frequently experienced agricultural droughts that have significantly impacted the economy and the ecological environment. The aim of this study is to examine the spatial and temporal fluctuations of agricultural drought in southwest China, as well as the contributing factors, by utilizing the Budyko model in tandem with The Standardized Soil Moisture Index (SSMI). The study identified four areas in Southwest China with a high incidence of agricultural drought. Yunnan and the Sichuan-Chongqing border regions had 10% of months with drought, while Guangxi and Guizhou had around 8% of months with drought. The droughts in these regions exhibit distinct seasonal characteristics, with Yunnan experiencing significantly higher drought frequency than other periods from January to June, while Guizhou and other areas are prone to severe droughts in summer and autumn. Using the Budyko model to characterize the water and energy balances of different arid regions in Southwest China, it can be observed that the water balances of Yunnan, Sichuan, and Chongqing are more vulnerable to moisture limitation. This is evidenced by the concentration of more Budyko points on the right-hand side of PET/P=1 in both regions. Conversely, the water balances of Guizhou and Guangxi are more stable, as demonstrated by the greater concentration of Budyko points in the vicinity of PET/P=1 in both regions. These findings, when combined with drought monitoring results, indicate that in drier years in Southwest China, the proportion of regional evapotranspiration in the water cycle increases while the proportion of available water resources decreases accordingly. After comparing the causes of drought and non-drought years, it is found that the average rainfall in southwest China is approximately 30% below normal during drought years, and the temperature is 1-2% higher than normal. This phenomenon is most noticeable during the spring and winter months. Additionally, vegetation transpiration is about 10% greater than normal during dry years in Southwest China, and soil evaporation increases by about 5% during the summer and autumn months compared to normal conditions.”

Q3: In the Introduction: The authors should point out the research goal of this manuscript and emphasize its research significance. Moreover, the authors should introduce the implications of previous related published research from a broader perspective.

Response:

Thank you very much for your suggestion. We have revised the introduction of the article based on the information you provided and have added relevant references.

“  Drought is a pervasive natural disaster worldwide, with over half of the earth's land surface facing the threat of drought(Allen et al., 2010). Unfortunately, the frequency, intensity, and duration of droughts have been on the rise in recent years due to the escalating impacts of climate change(Dai, 2013). Furthermore, the impact of droughts has a significant cumulative effect, which has emerged as a crucial factor hindering the sustainable development of economies and societies in the future (Alam et al., 2021). Of all the sectors impacted by drought, agriculture was the most affected and faced direct consequences (Wu et al., 2011). Agricultural drought is responsible for more than 50% of the total damage caused by drought (Wu et al., 2021). The stability of society and a country's food security are closely linked to the agricultural sector. Therefore, it has become an urgent issue to conduct scientific analysis and research on agricultural drought (Liu et al., 2021, Shahzaman et al., 2021).

The World Meteorological Organization has classified drought into four distinct categories: meteorological drought, hydrological drought, agricultural drought, and water management drought (Mishra et al., 2010). The drought index has become a crucial method for monitoring and evaluating drought (Mendicino et al., 2008). Approximately 55 drought indices are widely used for drought monitoring and analysis, and they are roughly categorized into meteorological, hydrological, and agricultural drought indices (Keyantash et al., 2002). The primary input parameters for meteorological drought indexes are typically precipitation and temperature. One widely used index is the Standardized Precipitation Evapotranspiration Index (SPEI) (Vicente-Serrano et al., 2010), Palmer Drought Severity Index (PDSI) (Dai et al., 2004), and Standardized Precipitation Index (SPI) (Guttman, 1999). The hydrological drought indexes typically use runoff, groundwater, and other core input parameters. An example is the Surface Water Supply Index (SWSI) (Garen, 1993) and Streamflow Drought Index (SDI)(Du et al., 2013). The primary focus of agricultural drought indexes is soil moisture and vegetation ecological data, which are crucial for monitoring and analyzing agricultural droughts. Commonly used agricultural drought indices include the Standardized Soil Moisture Drought Index (SSMI) (Xu et al., 2018), the Soil Moisture Anomaly Percentage Index (SMAPI) (Wu et al., 2011), and the Evapotranspiration Deficit Index (ETDI) (Narasimhan et al., 2005). Additionally, the vegetation condition index (VCI)(Shahzaman et al., 2021) is frequently employed to monitor agricultural drought. In this study, we have opted to use SSMI as the monitoring indicator for agricultural drought and analyze its temporal and spatial variations in southwest China from 2005 to 2020 using SSMI.

Many researchers have conducted studies on monitoring agricultural drought, assessing its intensity, predicting it, and identifying the drivers behind it. Wilhelmi (Wilhelmi et al., 2002) proposed a spatial, GIS-based approach to assess the vulnerability of Nebraska to agricultural drought. They assessed the physical and social factors that determine agricultural drought vulnerability and demonstrated that the most vulnerable areas in the study area were non-irrigated drylands and rangelands on sandy soils. YuZhang (Zhang et al., 2021) utilized a meta-Gaussian model to forecast spring and summer agricultural droughts in China, taking into account soil moisture, Southern Oscillation, and other influencing factors. The findings indicated that the El Niño-Southern Oscillation (ENSO) can offer valuable early warning information for predicting agricultural droughts. Rhee and colleagues (2010) developed a new drought index called the Scaled Drought Condition Index (SDCI) for monitoring agricultural drought using multi-sensor data in both dry and wet regions. The results of their study showed that SDCI was effective in monitoring the spatial and temporal dynamics of agricultural drought in arid and humid areas (Rhee et al., 2010). Agricultural drought has been the subject of numerous studies conducted in various countries (Ajaz et al., 2019, Mannocchi et al., 2004, Ullah et al., 2023, Zeng et al., 2019, Zhang et al., 2021). However, regional agricultural drought drivers have rarely been analysed from a water balance perspective.

The Budyko model is widely recognized as the dominant international theoretical framework for investigating the climatic-hydrological interdependencies of watersheds (Han et al., 2021). The classical quantitative theory of the Budyko model is based on the climatic aridity index and the mean hydrological flux of the multi-year average state. Its main objective is to describe the relationship between the regional long-term water balance, which is limited by energy and water (Sposito, 2017). The Budyko model is widely applied in research on regional water balance and other aspects. For example, Maurer and colleagues utilized the Budyko model to investigate the impact of drought on the water balance in California. The findings revealed that fluctuations in precipitation and temperature across years had a significant influence on absolute runoff changes (Maurer et al., 2022) . Huang (Huang et al., 2017) assembled the Budyko model with the standardized precipitation index (SPI) and the standardized runoff index (SSI) to analyse the factors influencing the propagation of meteorological drought to hydrological drought in the Weihe River basin, China. The findings indicated that the transmission of meteorological drought to hydrological drought exhibited notable seasonal features, and the duration of the transmission was positively associated with the parameters of the Fu equation of the Budyko model. The Budyko model functions as a conceptual framework that offers a broad comprehension of the connections between regional water and energy balance elements(Maurer et al., 2022). It achieves this without requiring high-resolution data or numerous parameters. By considering evapotranspiration, it factors in the non-linear correlation between precipitation and available water across diverse climatic circumstances(Hrachowitz et al., 2017). The applicability of the Budyko model has been enhanced by incorporating parameters that enable adjustments to the deviations resulting from diverse lower bedding surfaces. These parameters have been introduced into the equations governing the Budyko framework.

Southwest China is the main distribution area of the karst landscape in China, and the ecological environment is fragile and vulnerable to extreme disasters (Lin et al., 2015). The frequency and intensity of droughts in Southwest China have increased significantly in recent years, leading to adverse effects on the ecological environment, people's lives, and socioeconomic development (Hao et al., 2015). The existing studies related to drought in Southwest China mostly focus on drought monitoring, while research on the drivers of agricultural drought from the water balance perspective is relatively limited. Understanding the drivers of drought is crucial for developing effective responses, and investigating the drivers of agricultural drought can provide a theoretical basis for drought prediction and response (Christian et al., 2021). This study is to analyse the 2000-2020 agricultural drought in Southwest China using the Budyko model combined with the drought index SSMI and to analyse the impact of different drivers on the regional agricultural drought in terms of water and energy balance. The Standardized Soil Moisture Drought Index (SSMI) was calculated using GLDAS soil moisture data to assess agricultural drought in the region. The water balance characteristics of Southwest China were explored using the Budyko model, and the changes in water balance components between consecutive drought years in the region and 2000-2020 were compared and analysed to investigate the impact of different drivers on agricultural drought. The results of this study could provide valuable insights for developing scientific responses to future droughts in Southwest China

Q3: What are the main aims and motivations for using different reanalysis datasets instead of observed datasets, which is unclear and even not seen as the scientific reason in the entire manuscript?

Response:

First of all, thank you very much for your comments and here is our explanation of the issue.

“In this study, our main data came from GLDAS data, which included soil moisture, rainfall, temperature and evapotranspiration. During the last submission process, some reviewers raised some questions about the time span of the drought monitoring, so we chose a longer span of soil moisture provided by FLDAS as supplementary data. This section is only a supplement to the results of the agricultural drought monitoring in this paper to show the reasonableness of the monitoring results.”
“At present, there is an obvious lack of data space in the remote sensing observation data set of soil moisture in southwest China, which may lead to inaccurate research results. Therefore, remote sensing data are not selected for related research in this experiment. As shown in the following figure, the soil moisture data of ESA_CCI has some data defects in space.”

Figure 1 Monthly soil moisture changes of ESA_CCI in southwest China from January to April 2005.

Q4: I feel the authors need to provide some discussion on the limitation of the study in the discussion section and what needs to be done to address those. For example; a comprehensive analysis of climate sensitivity, given the datasets (that are used in the study) has a significant uncertainty amongst their models. Although I see authors have mentioned this in their manuscript, it still needs more elaborated discussion. Finally, discuss some lights on the physical mechanism on which the dryness/wetness is increasing over the regions much-with the help of some recent literature.

Response:

Thank you very much for your suggestion. We have added relevant explanations and quoted relevant recommended documents.

“This research analyzed agricultural drought in Southwest China over the past 20 years by utilizing GLDAS soil moisture data and the SSMI drought index. The spatial distribution of agricultural drought was examined, leading to the identification of four high-incidence areas: Yunnan, Guizhou, Guangxi, and the border between Sichuan and Chongqing. Agricultural drought was more frequent in temperate and tropical regions than in boreal regions. The Budyko model was used to analyze water balance characteristics in the high-drought incidence areas, revealing that limited water balance was a significant factor contributing to the frequency of agricultural drought in Yunnan and the Sichuan-Chongqing border areas. In contrast, the water balance in Guangxi and Guizhou was more stable. Finally, the study analyzed the impact of different driving factors on agricultural drought and found that decreased precipitation and increased temperature can cause drought, while vigorous vegetation transpiration and soil evaporation can exacerbate it.

The present research has limitations. Firstly, the agricultural drought index SSMI is a single-factor index that is easy to calculate and compare across time and space. However, it does not capture the multi-scale characteristics of agricultural drought, leading to more errors in drought monitoring results. In the follow-up study, we will analyze the agricultural drought in southwest China using multi-indexes. This will help to improve the accuracy of the monitoring results. Furthermore, one of the major shortcomings of the experiment is that surface runoff and groundwater are not considered in this study. In follow-up research, it is planned to use machine learning in combination with the Budyko model to comprehensively consider the influence of natural and perceived factors on drought. The study mainly focused on long-term monitoring and drivers of agricultural drought, with limited investigation of spatial distribution and changes. Besides its simplicity, the Budyko model has certain limitations. It overlooks a significant portion of the water and energy cycle processes in its computations, which can lead to inaccuracies in the Budyko results. This is due to the fact that it is less complex than modern land surface water and energy balance models. This problem can be improved by using a more complete land surface process model (Community Land Model, CLM).

Despite these limitations, the study analyzed the impact of driving factors on agricultural drought in Southwest China from a water and energy balance. It used the agricultural drought monitoring index and Budyko model to quantitatively analyze water balance characteristics in drought-prone areas. The results provide a theoretical foundation for predicting and responding to future agricultural droughts in Southwest China. The methodological framework used in this study can also be applied to other types of droughts for fundamental analysis.

The results of this study on the temporal trends as well as seasonal variations of agricultural droughts in southwest China are consistent with the results of previous studies and provide side evidence of the applicability of SSMI in southwest China (Bingfang, 2010, Hao et al., 2015). Most previous studies on drought frequency in southwest China have focused on small areas, such as Yunnan Province or Guizhou Province, while relatively few statistics on drought frequency at large spatial scales have been conducted for the whole southwest China (Cheng et al., 2020). Previous studies on the drivers of drought in southwest China have mostly been analyzed from the perspective of atmospheric circulation (El Niño, Southern Oscillation, etc.) (Zhang et al., 2013), but this study analyses the driving mechanisms of agricultural drought in southwest China from the perspective of water balance. The present study also finds that agricultural drought in Southwest China is not only related to precipitation and temperature, but also to regional evapotranspiration. The results of the current study are consistent with those of Mas et al, which again validates the plausibility of the current results (Maes et al., 2012, Teuling et al., 2013). From the perspective of the physical mechanism of drought, the decrease of rainfall and the increase of evapotranspiration may be important inducing factors for the occurrence of drought(Mie Sein et al., 2021, Sein et al., 2022, Ullah et al., 2023). In addition, human activities have both increased and alleviated the drought(Iyakaremye et al., 2021). The relationship between rainfall and evapotranspiration and the calculation of surface parameters have been the focus of most previous studies on Budyko. For example, Gerrits (Gerrits et al., 2009) et al. used Budyko model to analyses the relationship between rainfall and evapotranspiration. Bai (Bai et al., 2020) et al. used multiple linear regression model and artificial neural network model to calculate Budyko model parameters for Chinese region. This study applies the Budyko model to the study of drought drivers. It quantifies the drivers of drought in southwest China by combining water balance principles. This study analyses the impact of evapotranspiration components on drought, unlike previous studies (Maurer et al., 2022).”

Q5: Conclusions are not clarified. Besides, these findings cannot be well supported by the data and the results.

Response:

Thank you very much for your opinion. We have adjusted the conclusion.

“The severity, coverage, and duration of agricultural drought in Southwest China from 2000to 2020 exhibit obvious spatiotemporal characteristics. The severity of drought in the agricultural sector in Southwest China from 2000 to 2014 showed a significant increase, while the frequency and duration of agricultural drought in the re-gion decreased significantly from 2015 to 2020. Based on the frequency of drought, four high-frequency regions were identified in Southwest China, namely Yunnan, the Sichuan-Chongqing border region, Guangxi, and Guizhou. The frequency of moderate to severe drought in the northwestern Sichuan region was 4-5%, significantly lower than in other regions. Overall, agricultural drought in Southwest China showed a spatial pattern of decreasing severity from the southwest to the northeast. The four high-frequency regions of agricultural drought in Southwest China exhibit obvious seasonal characteristics, with Yunnan being prone to drought in the spring and winter seasons, and the frequency of extreme drought in the winter and spring seasons is around 10%. The Sichuan-Chongqing border region, Guizhou, and Guangxi are prone to moderate drought in the spring and winter seasons, and severe and extreme droughts in the summer and autumn seasons. The frequency of extreme drought in the summer and autumn seasons in the Sichuan-Chongqing border region is about 6%, while that in Guangxi and Guizhou is about 4%.

The Budyko model was utilized to analyze the water balance characteristics of the four drought-prone regions. The results showed that the Budyko points in Yunnan and the Sichuan-Chongqing border areas were located more towards the upper right of the coordinate system (Figures 6 and 7), indicating that water availability was more limited in these areas. Concurrently, the ratio of potential evapotranspiration to precipitation in Guizhou and Guangxi was closer to 1.00. In other words, the water balance in Guangxi and Guizhou was more stable, which also explains why droughts occur more frequently in Yunnan and the Sichuan-Chongqing region than in Guangxi and Gui-zhou. By analyzing the results of single-year regional Budyko trajectories with regional drought trends, it was found that the more severe the agricultural drought, the higher the percentage of evapotranspiration and the lower the percentage of available water in the components of the water balance. This suggests that the soil moisture is lower in severe drought conditions.

By comparing the changes in driving factors between consecutive drought years and normal years, it was found that in the southwest region of China, consecutive drought years had significantly lower rainfall and higher temperatures. In addition, during the spring and winter seasons of drought years, vegetation transpiration was very active, while during the summer and autumn seasons, vegetation transpiration and soil evaporation were significantly higher than in non-drought years.”

[1]       Ajaz A, Taghvaeian S, Khand K ,et al. Development and Evaluation of an Agricultural Drought Index by Harnessing Soil Moisture and Weather Data [J]. Water, 2019, 11(7).

[2]       Alam I, Otani S, Majbauddin A ,et al. The Effects of Drought Severity and Its Aftereffects on Mortality in Bangladesh [J]. Yonago Acta Med, 2021, 64(3): 292-302.

[3]       Allen C D, Macalady A K, Chenchouni H ,et al. A global overview of drought and heat-induced tree mortality reveals emerging climate change risks for forests [J]. Forest Ecology and Management, 2010, 259(4): 660-84.

[4]       Bingfang L Q Y N Z F C S W. Drought Monitoring and Its Impacts Assessment in Southwest China Using Remote Sensing in the Spring of 2010 [J]. ACTA GEOGRAPHICA SINICA, 2010, (07): 771-80.

[5]       Cheng Q, Gao L, Zhong F ,et al. Spatiotemporal variations of drought in the Yunnan-Guizhou Plateau, southwest China, during 1960–2013 and their association with large-scale circulations and historical records [J]. 2020, 112: 106041.

[6]       Christian J I, Basara J B, Hunt E D ,et al. Global distribution, trends, and drivers of flash drought occurrence [J]. Nature Communications, 2021, 12(1): 6330.

[7]       Dai A, Trenberth K E, Qian T T. A global dataset of Palmer Drought Severity Index for 1870-2002: Relationship with soil moisture and effects of surface warming [J]. Journal of Hydrometeorology, 2004, 5(6): 1117-30.

[8]       Dai A G. Increasing drought under global warming in observations and models [J]. Nature Climate Change, 2013, 3(1): 52-8.

[9]       Du L T, Tian Q J, Yu T ,et al. A comprehensive drought monitoring method integrating MODIS and TRMM data [J]. International Journal of Applied Earth Observation and Geoinformation, 2013, 23: 245-53.

[10]     Garen D C. REVISED SURFACE-WATER SUPPLY INDEX FOR WESTERN UNITED-STATES [J]. Journal of Water Resources Planning and Management-Asce, 1993, 119(4): 437-554.

[11]     Guttman N B. Accepting the standardized precipitation index: A calculation algorithm [J]. Journal of the American Water Resources Association, 1999, 35(2): 311-22.

[12]     Han P-F, Istanbulluoglu E, Wan L ,et al. A New Hydrologic Sensitivity Framework for Unsteady-State Responses to Climate Change and Its Application to Catchments With Croplands in Illinois [J]. 2021, 57(8): e2020WR027762.

[13]     Hao C, Zhang J, Yao F. Combination of multi-sensor remote sensing data for drought monitoring over Southwest China [J]. International Journal of Applied Earth Observation and Geoinformation, 2015, 35: 270-83.

[14]     Hrachowitz M, Clark M P. HESS Opinions: The complementary merits of competing modelling philosophies in hydrology [J]. Hydrol Earth Syst Sci, 2017, 21(8): 3953-73.

[15]     Huang S, Li P, Huang Q ,et al. The propagation from meteorological to hydrological drought and its potential influence factors [J]. J Hydrol, 2017, 547: 184-95.

[16]     Iyakaremye V, Zeng G, Yang X ,et al. Increased high-temperature extremes and associated population exposure in Africa by the mid-21st century [J]. Science of The Total Environment, 2021, 790: 148162.

[17]     Keyantash J, Dracup J A. The quantification of drought: An evaluation of drought indices [J]. Bulletin of the American Meteorological Society, 2002, 83(8): 1167-80.

[18]     Lin W, Wen C, Wen Z ,et al. Drought in Southwest China: a review [J]. Atmospheric Oceanic Science Letters, 2015, 8(6): 339-44.

[19]     Liu X, Guo P, Tan Q ,et al. Drought disaster risk management based on optimal allocation of water resources [J]. Natural Hazards, 2021, 108(1): 285-308.

[20]     Maes W, Steppe K. Estimating evapotranspiration and drought stress with ground-based thermal remote sensing in agriculture: a review [J]. Journal of experimental botany, 2012, 63(13): 4671-712.

[21]     Mannocchi F, Francesca T, Vergni L. Agricultural drought: Indices, definition and analysis [J]. IAHS-AISH Publication, 2004: 246-54.

[22]     Maurer T, Avanzi F, Glaser S D ,et al. Drivers of drought-induced shifts in the water balance through a Budyko approach [J]. Hydrol Earth Syst Sci, 2022, 26(3): 589-607.

[23]     Maurer T, Avanzi F, Glaser S D ,et al. Drivers of drought-induced shifts in the water balance through a Budyko approach [J]. Hydrol Earth Syst Sci, 2022, 26(3): 589-607.

[24]     Mendicino G, Senatore A, Versace P. A Groundwater Resource Index (GRI) for drought monitoring and forecasting in a mediterranean climate [J]. J Hydrol, 2008, 357(3): 282-302.

[25]     Mie Sein Z M, Ullah I, Saleem F ,et al. Interdecadal Variability in Myanmar Rainfall in the Monsoon Season (May–October) Using Eigen Methods [J]. 2021, 13(5): 729.

[26]     Mishra A K, Singh V P. A review of drought concepts [J]. J Hydrol, 2010, 391(1-2): 204-16.

[27]     Narasimhan B, Srinivasan R. Development and evaluation of Soil Moisture Deficit Index (SMDI) and Evapotranspiration Deficit Index (ETDI) for agricultural drought monitoring [J]. Agricultural and Forest Meteorology, 2005, 133(1-4): 69-88.

[28]     Rhee J, Im J, Carbone G J. Monitoring agricultural drought for arid and humid regions using multi-sensor remote sensing data [J]. Remote Sensing of Environment, 2010, 114(12): 2875-87.

[29]     Sein Z M M, Zhi X, Ullah I ,et al. Recent variability of sub-seasonal monsoon precipitation and its potential drivers in Myanmar using in-situ observation during 1981–2020 [J]. International Journal of Climatology, 2022, 42(6): 3341-59.

[30]     Shahzaman M, Zhu W, Bilal M ,et al. Remote Sensing Indices for Spatial Monitoring of Agricultural Drought in South Asian Countries [J]. 2021, 13(11): 2059.

[31]     Shahzaman M, Zhu W, Ullah I ,et al. Comparison of Multi-Year Reanalysis, Models, and Satellite Remote Sensing Products for Agricultural Drought Monitoring over South Asian Countries [J]. 2021, 13(16): 3294.

[32]     Sposito G. Understanding the Budyko equation [J]. Water, 2017, 9(4): 236.

[33]     Teuling A J, Van Loon A F, Seneviratne S I ,et al. Evapotranspiration amplifies European summer drought [J]. Geophysical Research Letters, 2013, 40(10): 2071-5.

[34]     Ullah R, Khan J, Ullah I ,et al. Assessing Impacts of Flood and Drought over the Punjab Region of Pakistan Using Multi-Satellite Data Products [J]. 2023, 15(6): 1484.

[35]     Ullah R, Khan J, Ullah I ,et al. Investigating Drought and Flood Evolution Based on Remote Sensing Data Products over the Punjab Region in Pakistan [J]. 2023, 15(6): 1680.

[36]     Vicente-Serrano S M, Begueria S, Lopez-Moreno J I. A Multiscalar Drought Index Sensitive to Global Warming: The Standardized Precipitation Evapotranspiration Index [J]. Journal of Climate, 2010, 23(7): 1696-718.

[37]     Wilhelmi O V, Wilhite D A. Assessing Vulnerability to Agricultural Drought: A Nebraska Case Study [J]. Natural Hazards, 2002, 25(1): 37-58.

[38]     Wu D, Li Y A, Kong H ,et al. Scientometric Analysis-Based Review for Drought Modelling, Indices, Types, and Forecasting Especially in Asia [J]. Water, 2021, 13(18): 2593.

[39]     Wu Z Y, Lu G H, Wen L ,et al. Reconstructing and analyzing China's fifty-nine year (1951-2009) drought history using hydrological model simulation [J]. Hydrol Earth Syst Sci, 2011, 15(9): 2881-94.

[40]     Xu Y, Wang L, Ross K W ,et al. Standardized Soil Moisture Index for Drought Monitoring Based on SMAP Observations and 36 Years of NLDAS Data: A Case Study in the Southeast United States [J]. Remote Sens (Basel), 2018, 10(2): 134-78.

[41]     Zeng Z, Wu W, Li Z ,et al. Agricultural Drought Risk Assessment in Southwest China [J]. 2019, 11(5): 1064.

[42]     Zhang W, Jin F-F, Zhao J-X ,et al. The possible influence of a nonconventional El Niño on the severe autumn drought of 2009 in Southwest China [J]. 2013, 26(21): 8392-405.

[43]     Zhang Y, Hao Z, Feng S. Agricultural drought prediction in China based on drought propagation and large-scale drivers [J]. Agricultural Water Management, 2021, 255: 107028.

[44]     Zhang Y, Hao Z, Feng S ,et al. Agricultural drought prediction in China based on drought propagation and large-scale drivers [J]. Agricultural Water Management, 2021, 255: 107028.

Round 2

Reviewer 3 Report (New Reviewer)

The authors went through a great length and improved the revised manuscript as per suggestions and comments. I must applaud the authors for possible publications in the esteemed journal of MDPI “Remote Sensing”. However, I have still minor concerns regarding the title and think it is not suited to the objectives of the current work. My title suggestion is “Attribution of Extreme Drought Events and Associated Physical Drivers across Southwest China using the Budyko Framework”, if accepted for authors.

Minor editing of English language required

Author Response

Response to reviewer

Dear reviewer:

Q1:The authors went through a great length and improved the revised manuscript as per suggestions and comments. I must applaud the authors for possible publications in the esteemed journal of MDPI “Remote Sensing”. However, I have still minor concerns regarding the title and think it is not suited to the objectives of the current work. My title suggestion is “Attribution of Extreme Drought Events and Associated Physical Drivers across Southwest China using the Budyko Framework”, if accepted for authors.

Response:

Thank you very much for your suggestion. We have fully considered your suggestion and accepted your modification suggestions. We have made modifications to the title of this article.

“Attribution of Extreme Drought Events and Associated Physical Drivers across Southwest China using the Budyko Framework”

This manuscript is a resubmission of an earlier submission. The following is a list of the peer review reports and author responses from that submission.

Round 1

Reviewer 1 Report

Dear Authors,

The paper is interesting since the topic fails within the the scope of the journal and it is always valued to bring new remote sensing dataset in the every day life of science. My previous comment has not been fully responded since 20 years of dataset is being used. This is a major discrepancy in comparison with habitual drought practise research. 

To be more specific in my latest paper (Tegos et al 2023) I used one of most globally well known meteorological network for calculating potential evapotranspiration. Among 50 gauge station I identified only 6 stations for which a period of 30 years with full monthly records of rainfall and PET is covered.

I fully understand the problem of the long term record, however, you will now need to collect data from gauge stations with at least 30 years and provide more analysis in quantifying the drought severity comparing to remote sensing dataset. That will benefit your paper and will be in line with your discussion sections where similar concerns are highlighted.

The paper without this extension setting up above can not be accepted.

Reference

Tegos A, Stefanidis S, Cody J, Koutsoyiannis D. On the Sensitivity of Standardized-Precipitation-Evapotranspiration and Aridity Indexes Using Alternative Potential Evapotranspiration Models. Hydrology. 2023; 10(3):64. https://doi.org/10.3390/hydrology10030064

Reviewer 2 Report

Review of the manuscript “Characteristics and Drivers of Extreme Drought in Southwest 2 China under Budyko Framework” (remotesensing-2377401). This manuscript tries to utilize the Standardized Soil Moisture Index (SSMI) and the Budyko model to investigate agricultural drought in southwest China. Although the manuscript fits the scope of the RS journal and the amount of the work is enough, its scientific contribution to the analysis of characteristics and drivers of agricultural drought needs to be explained much more clearly and convincingly. More detailed comments and suggestions are listed as follows:

1. The abstract needs more qualitative information derived from the study. The authors need to add more quantitative information in the abstract.

2. From my point of view, the methodology of the Budyko model has not been innovatively developed by this study. This manuscript merely presented a package incorporating associated methods and tools.

3. in section 3.3. Analysis of Agricultural Drought Drivers, agricultural drought is proposed based on meteorological data, soil moisture, etc, that is when considering the driving factors of agricultural drought, this manuscript merely considers the meteorological data itself. The drivers of agricultural drought should contain climate factors, human factors, vegetation factors, underground water, land cover, etc.

4. From my point of view, there is no real discussion. The Discussion Sections failed to engage with the wider readership of this international and interdisciplinary journal. Furthermore, there is no comparison of the procedures and results of different authors, especially for the Budyko model used in this manuscript.

no comments